# Validation of a Classification System for Optimal Application of Debridement, Antibiotics, and Implant Retention in Prosthetic Joint Infections following Total Knee Arthroplasty: A Retrospective Review

**DOI:** 10.3390/antibiotics13010048

**Published:** 2024-01-04

**Authors:** Joyee Tseng, Victoria Oladipo, Siddhartha Dandamudi, Conor M. Jones, Brett R. Levine

**Affiliations:** Rush University Medical Center, Department of Orthopaedic Surgery, Chicago, IL 60612, USA; tseng.joyee@gmail.com (J.T.); va.oladipo@gmail.com (V.O.); levine.research@rushortho.com (S.D.); conor_m_jones@rush.edu (C.M.J.)

**Keywords:** prosthetic joint infection, total knee arthroplasty, debridement and implant retention, classification

## Abstract

Introduction: Periprosthetic joint infection (PJI) remains a serious complication after total knee arthroplasty (TKA). While debridement, antibiotics, and implant retention (DAIR) are considered for acute PJI, success rates vary. This study aims to assess a new scoring system’s accuracy in predicting DAIR success. Methods: 119 TKA patients (2008–2019) diagnosed with PJI who underwent DAIR were included for analysis. Data were collected on demographics, laboratory values, and clinical outcomes. This was used for validation of the novel classification system consisting of PJI acuteness, microorganism classification, and host health for DAIR indication. Statistical analysis was carried out using SPSS programming. Results: Mean follow-up was 2.5 years with an average age of 65.5 ± 9.1 years, BMI of 31.9 ± 6.2 kg/m^2^, and CCI of 3.04 ± 1.8. Successful infection eradication occurred in 75.6% of patients. The classification system demonstrated 61.1% sensitivity, 72.4% specificity, and 87.3% positive predictive value (PPV) when the DAIR cutoff was a score less than 6. For a cutoff of less than 8, sensitivity was 100%, specificity was 37.9%, and PPV was 83.3%. Conclusions: To date, no consensus exists on a classification system predicting DAIR success. This novel scoring system, with high PPV, shows promise. Further refinement is essential for enhanced predictive accuracy.

## 1. Introduction

Periprosthetic joint infection (PJI) is a postoperative complication in up to 2% of primary total knee arthroplasties (TKAs) [1]. The rate of primary TKAs is expected to increase by 43% by 2050 [2]. As TKAs continue to rise, the number of patients who will experience PJI will concomitantly increase. PJIs can be categorized as acute, late acute (hematogenous spread), or chronic. Acute PJI is defined as occurring within 4 weeks of the index procedure. Late acute hematogenous PJI is defined as acute symptoms of less than 3 or 4 weeks in a previously well-functioning prosthesis [3]. The typical treatment for acute and hematogenous PJIs is often debridement, antibiotics, and implant retention (DAIR) [4].

DAIR is an appealing option for acute PJI as it allows retention of the original implants, a relatively short recovery time, and reasonable infection eradication rates [5,6,7]. Early DAIR procedures are more effective in treating acute PJI as this may occur prior to the maturation of biofilm formation by the infecting bacteria [8]. Success rates of DAIR have been reported to range from 60–80% in multiple small studies [4,9]. Recently, McQuivey et al. described a double DAIR procedure with the use of high-dose antibiotics between stages of treatment with overall infection-control rates of 87% and 90% in two separate studies at an average of 3.5 years follow-up [7,10]. The timing of a DAIR procedure is critical, as studies have illustrated that the longer a patient has infectious symptoms, the more likely a DAIR will fail to eradicate the infection [11,12]. Patients with chronic PJI are typically not candidates for DAIR and follow the historical gold standard, a two-stage exchange procedure (75–95% effective) [13,14,15].

A tool that accurately predicts the success of DAIR prior to surgery would be incredibly helpful in these difficult cases to minimize the number of necessary surgeries and improve long-term outcomes. Risk factor scoring systems such as the KLIC score and the CRIME-80 have attempted to predict the success of DAIR for early PJI and late acute (hematogenous), respectively, by assessing host factors alongside lab values [8,16,17,18]. The KLIC score was predictive of failure in early PJI and the CRIME-80 score was predictive of failure for hematogenous infections [8,16,17,19]. On external validation, the KLIC score was shown to have an area under the receiver-operating curve (AUC) to be 0.64 (0.839 internal validation), and the CRIME-80 was shown to have an AUC of 0.61 leaving room for improvement and adaptations to these scoring systems to predict a successful DAIR [16,17,20,21,22,23].

Recently, a new, simplified scoring system was proposed at the Israeli Orthopedic Association (IOA) meeting in 2022. This novel scoring system considers three factors to determine if a DAIR procedure is indicated: the chronicity of the PJI, virulence of the bacteria, and the patient’s overall health. This study aims to retrospectively apply this scoring system to a cohort of TKA patients who underwent DAIR to assess the predictive accuracy of the classification system at our institution.

## 2. Results

A total of 119 patients were included based on criteria and had a mean follow-up of 2.5 years (0.11–10.7 years). Overall baseline characteristics consisted of 42% females, CCI of 3.04 ± 1.8, average age of 65.5 ± 9.1 years, and BMI of 31.9 ± 6.2 kg/m^2^. The cohort included 6.7% current smokers, 32.8% former smokers, and 60.5% never smokers (Table 1).

Successful eradication was found in 90 patients (75.6%). Those who did not achieve infection eradication had a significantly higher percentage of smokers (21% vs. 2%, ***p = 0.025***) and were younger (62.6 ± 8.1 vs. 66.5 ± 9.3 years, ***p = 0.034***) than patients who did. The patients who did not achieve infection eradication were noted to have a significantly higher preoperative CRP (197.9 ± 120.2 vs. 142.2 ± 98.1 years, ***p = 0.016***) and trend towards a higher ESR (70.5 ± 35.8 vs. 56.0 ± 35.5 years, *p = 0.067*) but similar albumin levels (*p = 0.542*) (Table 2). The most prevalent microorganism isolated in our cohort was MSSA (47.5%). Among the patients who did not successfully eradicate the infection after DAIR, MSSA was also the most prevalent microorganism at 40.7% with multi-organisms following next at 22.2%.

Considering the IOA classification subclassifies the indication for DAIR as indicated or feasible based on score cut-offs, we stratified accuracy using both thresholds of 6 and 8. The classification showed a sensitivity of 61.1% when a score of less than 6 was used as an indication for DAIR. The specificity in predicting when patients should not be indicated for DAIR with this cut-off was 72.4% as the classification system supported implant removal. The positive predictive value was 87.3% to show the probability of allowing for DAIR and achieving infection eradication. If the cut-off score was set to be less than 8, the classification system had a sensitivity rate of 100% but a specificity of 37.9% and a PPV of 83.3% (Table 3).

When stratifying patients based on a score of 6 or less for DAIR indication, those who correctly adhered to the classification exhibited no significant differences in BMI, smoking, age, CCI, ESR, CRP, or albumin compared to patients who deviated from the classification. However, a noticeable trend was observed among patients who deviated from the scoring system, showing higher ESR (*p = 0.07*) and lower albumin (*p = 0.07*). When utilizing a score of 8 or less for DAIR indication, patients who deviated demonstrated a statistically significant increase in CRP (211.6 ± 117.6 vs. 140.4 ± 98.0, ***p = 0.003***) and a trending decrease in age (62.7 ± 8 vs. 66.3 ± 9.4, *p = 0.07*) (Table 4).

## 3. Materials and Methods

Approval from the Institutional Review Board was acquired. This is a single-center, multi-surgeon, retrospective study reviewing patients from 2008 to January 2019. All primary TKA patients who were diagnosed with PJI and underwent a DAIR procedure with polyethylene liner exchange were identified. Due to the lack of availability and data collection dates, the 2013 Musculoskeletal Infection Society (MSIS) criteria was used instead of the contemporary 2018 MSIS or ICM criterion [24,25]. Patients were excluded if they were less than 18 years of age, were missing the necessary data to complete the classification system, or did not meet the MSIS criteria. A total of 119 patients (74%) were ultimately included after applying exclusion criteria to 161 identified patients.

Patient demographic information was collected including age, gender, BMI, smoking status, and comorbidities with the use of the Charlson Comorbidity Index (CCI). Additionally, pre-operative assessment of routine laboratory data such as, C-reactive protein (CRP), erythrocyte sedimentation rate (ESR), synovial white blood cell count (WBC), and synovial polymorphonuclear (PMN) percentages were captured for all patients. Details concerning prior surgical procedures, encompassing the index TKA, the time duration from the index TKA to the onset of infectious symptoms, previous revision surgeries, and subsequent surgical interventions were extracted from the electronic medical records. Infection eradication is defined by the Delphi criteria: (1) healed incision without signs of purulent drainage or sinus tract formation and no recurrence of the same organism; (2) no subsequent surgical intervention for infection after reimplantation surgery; and (3) no PJI-related mortality. Failure to eradicate infection was defined by the necessity of additional surgical intervention following the DAIR procedure secondary to persistent infection.

The proposed system from the IOA is displayed in Table 5. Patients are scored using infection acuteness, the offending bacteria, and relative host health. A score of 2 is assigned to any patient who has an acute or late acute hematogenous infection with less than 3 weeks of symptoms. A score of 4 is assigned to any patient with a late sub-acute infection of 3–4 weeks of symptoms. A score of 6 is assigned to any patient with a chronic infection. A score of 1 is assigned to a patient if their offending microorganism was a single bacterium that was non-resistant to antibiotics. A score of 2 is assigned to patients who had resistant microorganisms on culture (i.e., MRSA or MRSE). Resilient microorganisms on culture were defined as multiple offending bacteria, fungi, or vancomycin-resistant bacteria and patients received a score of 3. Host health was described using CCI score which classified patients into healthy when CCI < 2, mildly immunocompromised when CCI < 4, and severely compromised when CCI ≥ 4. Scores are given 1, 2, and 3, respectively. The classification says DAIR is indicated if a patient scores either a 4 or 5, feasible if a patient scores either 6 or 7, and contra-indicated in patients who score 8 or more. Adherence to the IOA classification was defined as success of DAIR below a cut-off score and failure of DAIR above the cut-off score. Deviation was defined as failure of DAIR in infection eradication below a cut-off score and success of DAIR above the cut-off score.

Statistical methods included two-tailed *t*-tests for individual cohort comparisons were calculated when the outcome of interest was quantitative. A *p*-value of <0.05 was considered statistically significant. For continuous variables, means were reported with standard deviations or ranges. Univariate analysis including sensitivity and specificity were calculated for evaluation of the classification system. This was all performed using Statistical Package for the Social Sciences programming (IBM SPSS, Version 29.0. Armonk, NY, USA: IBM Corp.).

## 4. Discussion

It is imperative to accurately recognize and manage PJI in patients who have undergone TKA to avoid treatment failure and patient morbidity. There is no doubt significant appeal to the DAIR procedure, as it is less time consuming, lower risk, easier to recover from, and reimburses favorably compared to removal of the implants and spacer placement [26]. However, selecting the most suitable candidates for DAIR continues to be a complex challenge, primarily due to the absence of universally validated and highly effective guidelines. The complexity of this task underscores the significance of surgeons making this nuanced decision and comprehending prognostic factors that may hinder treatment effectiveness. This study demonstrates the proposed system’s relative effectiveness in predicting DAIR success for patients with infected TKA.

The proposed system takes a holistic approach to PJI by assessing the acuteness of the infection, identifying the microorganism, and evaluating patient comorbidities to output into a single numerical value that assesses if DAIR is reasonable to eradicate the infection. Following the scoring system (Table 5), a cut-off score of 6 offered a high specificity (72.4%) which is helpful in ruling out DAIR for patients with PJI and high PPV (87.3%), which shows it can accurately indicate DAIR. When the cut-off score of 8 is used, the sensitivity increased from 61.1% to 100% which does make sense given it captures an increased number of patients indicated for a DAIR, and successfully eradicated the infection. With this cut-off score, there was also a high PPV of 83.3% indicating this classification has the potential to likely be correct.

The first parameter in the scoring system is the acuteness of the infection, which has the largest weight of the overall score. In 2019, the international consensus on orthopedic infections commented on successful DAIR being correlated with less than 1 week of symptom duration and the age of the implant is less than 15 days [27]. In addition, it has been shown that hematogenous infections are independent predictors of DAIR failure and success of DAIR is much higher in acute patients than late acute patients [8,12,28,29].

In the new system, patients are scored by accounting for both time from index surgery and the number of weeks of infectious symptoms prior to DAIR. The system correlates the time of infectious onset in acute late PJI patients directly to either a score of 2 or 4 to indicate that patients with longer periods of symptoms are not as great candidates for DAIR. It also excludes longstanding hematogenous spread PJI or true chronic infections by ensuring that patients with true chronic infections are not considered suitable candidates for DAIR by assigning 6 points immediately. When this is added to the other two categories, the patient will score a minimum of 8 and DAIR will be contraindicated. The novel system does an excellent job of not overpredicting the success of DAIR.

The second parameter in the classification system is scoring based on the microorganism causing the infection. It has been well established that the microorganism can change the outcome of a DAIR [4,8,30,31,32,33]. Patients with fungal, polymicrobial infections, and resistant bacteria (MRSA/MRSE) may be better off with a two-stage revision [11,15,27]. Staphylococcal infections are also a huge factor in the success or failure of treating PJIs. Various results have been published showing success rates of DAIR with staphylococcus aureus ranging from 13% to 90% [27,34]. Staphylococcal aureus infection may be an independent risk factor for DAIR failure as well [35,36]. Rudelli et al. show that multidrug-sensitive microorganisms have lower failure rates of DAIR, indicating that resistance to drugs in treatment can impact DAIR success [37]. Overall, the microorganism causing the infection should play a part in determining if a patient gets DAIR; however, the specific weight placed on the individual organism likely needs further investigation to improve the proposed classification system [37].

Among the patients with polymicrobial cultures, 60% failed to fully eradicate the infection with DAIR. These individuals, had the proposed classification system been in place, would have scored higher and been deemed unsuitable candidates for DAIR. Notably, the cohort exhibited a notable percentage of MSSA infections with a concurrent high rate of DAIR failure in these patients. Considering the choice of antibiotics can impact infectious outcomes, enhancing the scoring system by introducing a criterion or modifying the existing microorganism parameter may strengthen the scoring system.

The final parameter in the classification system is the overall health of the patient assessed by the CCI score [38]. It has been long thought that the host immune system and patients’ general health is a critical factor in contracting and/or the ability to PJI. Host-related factors such as rheumatoid arthritis, older age, liver cirrhosis, and immunosuppression have been shown to be associated with failure of DAIR [8,16,27]. Establishing a portion of the novel scoring system to assess patient health creates a holistic picture of the patient to predict DAIR success. Our cohort did show that smoking caused patients to deviate from the prediction of the score. Smoking was not factored into the classification system and may be a worthwhile addition to the host health parameter.

This study did not find BMI, ASA score, CCI, gender, albumin, or synovial WBC count to be associated with an increased risk of treatment failure. We did see a significantly higher CRP and trending higher ESR for patients with failure of infection eradication which aligns with the idea that the higher ESR and CRP may coincide with the severity of infection although these lab values are historically more useful in chronic PJI [39]. With further investigation, lab values may be valuable to add as additional categories into the proposed classification to increase the predictive value of the score. Evidence has shown us how lab values can play a role in predicting DAIR success via the KLIC and CRIME-80 scores [16,21,22,23].

There are some limitations noted in this study. Firstly, this research was conducted within a single institution. Although multiple surgeons were included in the data, surgeon bias regarding the appropriate use of DAIR may be present. Future direction may include expansion to other academic institutions to allow natural variance in practice to limit this confounding factor. This was also a retrospective study, which inherently has its own limitations, including incomplete data, but this was mitigated by properly chart reviewing all available electronic medical records on the patients.

While this novel system has been presented at IOA, it has not been published and therefore has not been validated by external data. Our goal was to assess the accuracy of the novel system in our institution’s database to determine the merit of this scoring system. At our institution, where we frequently handle complex infection revisions, the proposed system can potentially quickly identify suitable candidates for DAIR and offer a PPV in the success of a DAIR. While the score itself does not make the decision to perform surgery on a patient, it offers a valuable tool in stratifying patients’ risk of failure of DAIR, allowing a discussion and decision to be made by the surgeon and patient. This proposed system needs further robust validation as well as additional scoring points, but currently does offer a new meaningful outlook on optimizing predictions for treatment outcomes in PJI in patients who have undergone TKA.

## Figures and Tables

**Table 1 antibiotics-13-00048-t001:** Patient demographics. BMI, body mass index (kg/m^2^); CCI, Charlson Comorbidity Index; SD, Standard Deviation.

Variable		Mean	SD
Total Cohort (*n*)	119		
Age (Years)		65.54	9.12
BMI		31.91	6.21
Gender (Women, *n*, (%))	50, 42		
Smoking (*n*, (%))			
Current	8, 6.7		
Former	39, 32.8		
Never	72, 60.5		
CCI		3.04	1.82
Follow-up (years)		2.51	2.16

**Table 2 antibiotics-13-00048-t002:** Infection eradication cohort vs. failure of infection eradication cohort. BMI, body mass index (kg/m^2^); CCI, Charlson Comorbidity Index; ESR, erythrocyte sedimentation rate (mm/h); CRP, C-reactive protein (mg/L).

	Infection Eradication Cohort (*n* = 90) [Mean ± SD]	Failure of Infection Eradication (*n* = 29) [Mean ± SD]	*p*-Value
BMI	32.06 ± 5.94	31.46 ± 7.08	*p = 0.649*
Age (years)	66.50 ± 9.34	62.59 ± 8.14	** *p = 0.034* **
Current Smokers (%)	2 ± 1.5	21 ± 4.1	** *p = 0.025* **
CCI	3.02 ± 1.81	3.10 ± 1.92	*p = 0.836*
ESR	55.99 ± 35.84	70.52 ± 35.84	*p = 0.067*
CRP	142.20 ± 98.19	197.89 ± 120.25	** *p = 0.016* **
Albumin	3.29 ± 0.80	3.21 ± 0.51	*p = 0.542*

**Table 3 antibiotics-13-00048-t003:** Novel classification system reliability. PPV, Positive predictive Value.

Classification Cut-Off Score	Total Score	*n*	%	Sensitivity	Specificity	PPV
6				0.6111	0.7241	0.8730
	<6	63	48.09%			
	≥6	56	42.75%			
8				1.0000	0.3793	0.8333
	<8	108	82.44%			
	≥8	11	8.40%			

**Table 4 antibiotics-13-00048-t004:** Patients who adhered to the classification vs. patients who deviated from the classification. BMI, body mass index (kg/m^2^); CCI, Charlson Comorbidity Index; ESR, erythrocyte sedimentation rate (mm/h); CRP, C-reactive protein (mg/L).

	Adherence to Classification System (Cut-Off 6) [Mean ± SD]	Deviation from Classification System (Cut-Off 6) [Mean ± SD]	*p*-Value	Adherence to Classification System (Cut-Off 8) [Mean ± SD]	Deviation from Classification System (Cut-Off 8) [Mean ± SD]	*p*-Value
BMI	31.96 ± 6.27	31.73 ± 6.11	*p = 0.874*	31.89 ± 5.93	32.04 ± 7.25	*p = 0.914*
Age (years)	65.75 ± 9.31	64.64 ± 8.75	*p = 0.609*	66.34 ± 9.37	62.69 ± 8.00	*p = 0.073*
Current Smokers (%)	5 ± 2.2	14 ± 3.5	*p = 0.289*	4 ± 2.1	15 ± 3.7	*p = 0.151*
CCI	3.06 ± 1.84	2.95 ± 1.79	*p = 0.805*	3.11 ± 1.87	2.81 ± 1.67	*p = 0.406*
ESR	56.76 ± 56.40	72.84 ± 36.81	*p = 0.076*	57.27 ± 36.34	67.63 ± 34.17	*p = 0.213*
CRP	148.19 ± 103.12	191.39 ± 115.46	*p = 0.105*	140.39 ± 98.01	211.61 ± 117.59	** *p = 0.003* **
Albumin	3.33 ± 0.69	2.98 ± 0.87	*p = 0.076*	3.27 ± 0.79	3.26 ± 0.49	*p = 0.913*

**Table 5 antibiotics-13-00048-t005:** IOA Novel Classification System. Total score is summed from each parameter. MRSA—Methicillin-resistant Staphylococcus Aureus; MSSE—Methicillin-resistant Staphylococcus Epidermidis; MSSE—Methicillin-sensitive Staphylococcus Epidermidis; MSSA—Methicillin-sensitive Staphylococcus Aureus.

Parameter	Sub-Parameter	Definition/Examples	Score
1. Acuteness	Acute	Acute or late acute hematogenous infection: <3 weeks of symptoms after onset	2
	Sub-acute	Late sub-acute infection: 3–4 weeks of symptoms	4
	Chronic	Chronic infection: >4 weeks after onset of symptoms	6
2. Bacteria	Normal	Single bacteria, MSSE, MSSA	1
	Moderately Resilient Bacteria	MRSA, MRSE	2
	Resilient Bacteria	Vancomycin-resistant Enterococci, Polyorganism, Fungi	3
3. Host	Normal/Mildly Compromised	CCI (0–1)	1
	Moderately Compromised	CCI (2–3)	2
	Severely Compromised	CCI (>4)	3

## Data Availability

The data are available as follows: Consulting—Link, Exactcech, Zimmer Biomet; Royalties—Link, Human Kinetics, Elsevier, Wolters Kluwer; Committees—AAOS-Quality Committee; MAOA—Education Committee; Deputy Editor Arthroplasty Today.

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
