# Peer review of "Validation of a Classification System for Optimal Application of Debridement, Antibiotics, and Implant Retention in Prosthetic Joint Infections following Total Knee Arthroplasty: A Retrospective Review"

_antibiotics, 2024, doi:10.3390/antibiotics13010048_

Round 1
Reviewer 1 Report
Comments and Suggestions for Authors
This is a paper on validation of IOA socring system for DAIR in acute and late acute PJIs.
Overall excellent work. Only small comments:
1- Tittle: do no use DAIR in tittle.
2- Material and methods: in statistical section there is no mention of the p value used to consider statistical significance. I'm gessing P value<0.05?
3- Results: all p values should be written in italics in all text and tables. (for exemple: 197.9 ± 120.2 vs 142.2 ± 98.1 years, p = 0.016)
4- Tables: I recommend to mark in bold letters those p values that reach statistical significance. It makes is more easy to the reader.
5- Discussion: page 7 line 208--> et all. is missing ("Rudelli et al.") , also reference 37 is missing.
Author Response
We updated the title, clarified the p value levels and made sure the pvalues stand out appropriately. We updated reference 37 and added the authors.

Reviewer 2 Report
Comments and Suggestions for Authors
The paper presents a classification system of patients depending on age, stage of infection, infection organism and health for the application of DAIR in prosthetic joint infections.
It is well presented and easy to follow. Although there are drawbacks in the study, these are listed in the limitations and for future study purposes. For further research, it would be beneficial to study in a boarder population at a wide age groups and in multiple clinical settings.
Overall, the system is useful for an initial assessment of patients that would require DAIR.
Author Response
We updated all points of both reviewers to make this a stronger manuscript. See the attached letter for detailed points.
